# BDE-47, -99, -209 and Their Ternary Mixture Disrupt Glucose and Lipid Metabolism of Hepg2 Cells at Dietary Relevant Concentrations: Mechanistic Insight through Integrated Transcriptomics and Proteomics Analysis

**DOI:** 10.3390/ijms232214465

**Published:** 2022-11-21

**Authors:** Marialuisa Casella, Gabriele Lori, Lucia Coppola, Cinzia La Rocca, Sabrina Tait

**Affiliations:** 1Core Facilities, Istituto Superiore di Sanità, Viale Regina Elena 299, 00161 Rome, Italy; 2Center for Gender-Specific Medicine, Istituto Superiore di Sanità, Viale Regina Elena 299, 00161 Rome, Italy; 3Science Department, Università Degli Studi di Roma Tre, Viale Guglielmo Marconi 446, 00146 Rome, Italy; 4Department of Experimental Medicine, Sapienza University of Rome, Viale Regina Elena 324, 00161 Rome, Italy

**Keywords:** metabolic disruptors, food contaminants, mixtures, new approach methodologies, in vitro toxicology, mode of action, omics

## Abstract

Polybrominated diphenyl ethers (PBDEs) are persistent organic chemicals implied as flame retardants. Humans are mainly exposed to BDE-47, -99, and -209 congeners by diet. PBDEs are metabolic disruptors with the liver as the main target organ. To investigate their mode of action at a human-relevant concentration, we exposed HepG2 cells to these congeners and their mixture at 1 nM, analyzing their transcriptomic and proteomic profiles. KEGG pathways and GSEA Hallmarks enrichment analyses evidenced that BDE-47 disrupted the glucose metabolism and hypoxia pathway; all the congeners and the MIX affected lipid metabolism and signaling Hallmarks regulating metabolism as mTORC1 and PI3K/AKT/MTOR. These results were confirmed by glucose secretion depletion and increased lipid accumulation, especially in BDE-47 and -209 treated cells. These congeners also affected the EGFR/MAPK signaling; further, BDE-47 enriched the estrogen pathway. Interestingly, BDE-209 and the MIX increased ERα gene expression, whereas all the congeners and the MIX induced ERβ and PPARα. We also found that PBDEs modulated several lncRNAs and that HNRNAP1 represented a central hub in all the four interaction networks. Overall, the PBDEs investigated affected glucose and lipid metabolism with different underlying modes of action, as highlighted by the integrated omics analysis, at a dietary relevant concentration. These results may support the mechanism-based risk assessment of these compounds in relation to liver metabolism disruption.

## 1. Introduction

Polybrominated diphenyl ethers (PBDEs) are man-made chemicals with flame-retardant properties that are added to commercial products and materials such as textiles, plastics, electronic devices, household products, and furniture to reduce their flammability. Despite the EU ban and restrictions on commercial mixtures as PentaBDE, OctaBDE, and DecaBDE [1,2], they are still present as complex mixtures in the environment and the food chain due to their stability and persistence.

Humans are mainly exposed to PBDEs by diet, especially through the consumption of fatty foods such as meat and fish due to the high lipophilicity and bioaccumulation in fat matrices [3,4]. Additionally, dermal contact or oral dust ingestion are relevant routes of exposure since PBDEs are not chemically bound to the materials, thus leaching into indoor environments may easily occur. This raises concern for children due to their habit to crawl or put their hands to their mouth [4,5]. As a consequence, several epidemiological studies evidenced how children’s exposure levels are higher than in adults [6,7,8], with possible adverse outcomes for this vulnerable population group. Indeed, a decrease in intelligence quotient and an increase in hyperactive and aggressive behaviors [9,10], as well as memory impairment [11] are being increasingly associated with PBDE exposure in children.

Several observed effects may be ascribed to the endocrine disruptor (ED) properties of PBDE which can bind to the estrogen, progesterone, androgen, and glucocorticoid receptors [12,13]. However, their main characteristic is the structural similarity with thyroxine (T4) and triiodothyronine (T3) natural hormones with which they compete for the binding to thyroid receptors and transporters, leading to an imbalance in thyroid hormones homeostasis [14,15]. Since thyroid hormones (THs) control several physiological functions, including neurodevelopment [16], reproduction [17], and lipid metabolism [18], disruption of their circulating levels may have severe consequences such as hypothyroidism, [19,20], fertility deficiency [21,22] and reduced circulating lipid levels [23]. This last effect may be strictly related to hepatic steatosis and non-alcoholic fatty liver disease (NAFLD)-like phenotypes observed in rodent models developmentally exposed to BDE-47 [24]. In addition, an association between increased diabetes incidence and PBDE exposure has been also observed in recent human studies [24,25]. On the basis of this evidence, PBDEs are also considered metabolic disruptors or, more specifically, obesogens [26,27]. 

In a previous report, we investigated the relative toxicity potential of eight PBDE congeners considered by the European Food Safety Authority (EFSA) as the most relevant for dietary exposure i.e., BDE-28, 47, 99, 100, 153, 154, 183, and 209 [3]. By using two in vitro models, the human hepatocarcinoma (HepG2) and the human adenocarcinoma (DLD-1) cell lines, we integrated cytotoxicity, chemical properties, and exposure intake data, obtaining a toxicological ranking of the eight congeners. In particular, the three more abundant congeners had the relative toxicities scores: BDE-209 > 47 > 99 [28]. 

In the present study, we investigated more in detail their mechanisms of action in HepG2 cells by integrating transcriptomic and proteomic analyses upon treatment with single congeners or their mixture, at concentrations occurring in food [3,28]. In addition, to link omics data with phenotypic readouts, we analyzed glucose secretion, lipid accumulation, and gene expression of selected markers to better delineate their impact on liver metabolic functionality.

## 2. Results

### 2.1. Genes Differentially Expressed by PBDE Treatment

The congeners BDE-47, -99, and -209 modulated approximately the same number of genes in HepG2 cells (Table 1), with a prevalence of upregulation. The ternary mixture (MIX) of the PBDE congeners altered the expression of about three-fold the number of transcripts compared to single congeners, with a prevalent down-regulation.

Very few genes were commonly modulated by two or more congeners (Figure 1a), the major overlap occurring between BDE-99 and BDE-209 with 108 genes; 66 genes were modulated by all three congeners and only 9 were in common also with the MIX. Genes deregulated by MIX overlapped more with BDE-47 DEGs (44 genes) than with other DEG lists.

Along with protein-coding transcripts, a total of 156 lncRNAs (DELs) were identified within DEGs (Appendix A): 23 were specific for BDE-47 treatment, 16 for BDE-99, 11 for BDE-209, 71 for the MIX, and several were shared by two or more treatment conditions. In particular, seven DELs (*AC008966.1*, *AC093510.1*, *AL121899.1*, *CSNK1G2-AS1*, *CSTF3-DT*, *LINC01004*, *LOC114224*) were shared by BDE-47 and -99 conditions with same modulation, except *AC008966.1* which was repressed by BDE-47 and induced by BDE-99. Three DELs were induced by both BDE-47 and -209 (*AC245297.4*, *LINC00482*, *LINC01348*), seven DELs were induced by both BDE-99 and 209 (*AL359878.2*, *LINC00917*, *LINC02370*, *NEAT1*, *RMRP*, *SRRM2-AS1*, *ZNF503-AS2*), and seven DELs were induced by all the three single congeners (*AC004540.1*, *AC005523.2*, *AC020928.1*, *BAALC-AS1*, *HAGLR*, *SND1-IT1*, *TTTY18*). Only the lncRNA *AC010883.1* was shared among all the treatment conditions, being induced by the single congeners and repressed by the MIX.

### 2.2. Proteins Differentially Expressed by PBDE Treatment

Approximately the same numbers of differentially expressed proteins (DEPs) were observed in all four treatment conditions, including MIX, with a prevalence of expression induction (Table 1). A higher number of shared proteins was observed between BDE-47 and BDE-99 profiles (39 proteins), whereas the MIX profile had more similarities with BDE-209 DEPs (35 proteins) (Figure 1b). Only 12 proteins were modulated by all the single congeners and other 12 were modulated also by the MIX.

Very few overlaps were found comparing DEG and DEP lists for each condition (Appendix A), probably due to the different regulation and properties of genes and proteins or to differences in the two omics approaches.

Among the same features deregulated at the gene and protein expression level, 11 were identified for BDE-47 (Appendix A), but with a significant inverse correlation (Rho = −0.6399, *p* = 0.0340); 5 for both BDE-99 and -209 with no correlations; 38 for MIX with an inverse but not significant correlation (Rho = −0.5245, *p* = 0.0799).

### 2.3. KEGG Pathways and GSEA Hallmarks Enriched by PBDEs

The functional analysis was performed by combining the transcriptomic and proteomic datasets of each experimental condition to obtain a holistic view of the induced alterations. The two approaches used gave relevant combined evidence of the biological pathways mostly affected by PBDE congeners and mixture exposure on HepG2 cells.

By over-representation analysis performed with the ClueGO plugin of Cytoscape, we defined to what extent DEGs and DEPs differently contributed to KEGG pathway enrichment in the four experimental conditions, as shown in Appendix A and Appendix A.

BDE-47 enriched 26 pathways (Figure 2a), BDE-99 and -209 enriched 4 and 12 pathways, respectively, and MIX enriched 8 pathways. BDE-47 shared two pathways with BDE-99 (*gap junction* and *bacterial invasion of epithelial cells*) and the MIX (*protein processing in endoplasmatic reticulum* and *proteasome*), and three with BDE-209 (*fatty acid elongation*, *RNA transport,* and *prostate cancer*). 

BDE-47 was the only congener significantly affecting three pathways of carbohydrate metabolism, with *glycolysis/gluconeogenesis* being the top significant and featuring 14 modulated DEPs (12 out of 14 down-regulated), representing enzymes critically involved in the various steps of the metabolic pathway (Appendix A). The *HIF-1 signaling pathway*, the second most affected by BDE-47, shares 7 out of the 17 modulated DEGs/DEPs (mainly inhibited) with *glycolysis*, that is those regulating the anaerobic metabolism. 

BDE-47 also affected some endocrine system pathways such as those related to *estrogen signaling*, comprising nine induced keratins shared with the *Staphylococcus aureus infection* pathway; in addition, in the *thyroid hormone signaling pathways,* some upregulated DEGs and DEPs are shared with the *endocrine and other factor-regulated calcium reabsorption* and the *aldosterone-regulated sodium reabsorption* pathways. BDE-47 also affected pathways related to the immune system such as the *antigen processing and presentation* pathway, mainly featuring DEPs.

A core of DEGs and DEPs modulated by BDE-47 (Appendix A) represent relevant hubs linking several signaling pathways, i.e., the induced *EGFR, MAPK3, CDKN1B,* and MTOR, and the repressed *IL10, FOS, EGR1*, MAPK1, and STAT3. In particular, *EGFR, MAPK3,* and MAPK1 are involved in the *HIF-1, estrogen, FoxO,* and *Rap1 signaling pathways* as well as in *central carbon metabolism in cancer, prostate cancer, focal adhesion,* and *gap junction* pathways. 

BDE-99 enriched only four pathways, with *Salmonella infection* being the top significant, plus the *bacterial invasion of epithelial cells,* the *gap junction,* and the *spliceosome* pathways, mostly featuring induced DEPs and with no notably relevant hubs (Appendix A).

The most significant pathway affected by BDE-209 was the *adherens junction* followed by the *adrenergic signaling in cardiomyocytes,* mainly comprising down-regulated and upregulated DEPs, respectively (Appendix A). A central hub is the induced protein MAPK1 which is linked to the previously mentioned pathways and to *Chagas disease, dopaminergic synapse, long depression, AGE-RAGE signaling pathways in diabetic complications, prostate cancer, and IL-17 signaling pathway*. Further, similarly to what was observed for BDE-47, other relevant hubs are the upregulated *EGFR* and the repressed *FOS* and *EGR1*, as well as the induced PLCB1. Of relevance, the central role of the repressed *NFKB1*.

Eleven pathways were significantly affected by MIX, the top significant being the *olfactory transduction* pathway, mostly including repressed DEGs. Other relevant terms are *Parkinson* and *Huntington disease* pathways which share 26 features, mostly down-regulated DEGs (Appendix A). 

Most of the top hubs in the networks are featured in the *endocytosis* pathway such as the repressed *RHOA* and *SMAD3,* and four induced heat shock proteins (i.e., *HSPA1A, HSPA1B, HSPA8,* and HSPA2), being differently shared with the previous pathways as well as with *protein processing in endoplasmatic reticulum, legionellosis* and *cytokine–cytokine receptor interaction* pathways. 

The GSEA analysis, which considers the overall distribution of the entities (including magnitude of modulation and significance), yielded a limited list of significant Hallmark gene sets but more similarities among the four experimental conditions (Figure 2b). We found 9 significant Hallmarks for BDE-47, 6 for BDE-99, 8 for BDE-209, and 10 for MIX. 

*Glycolysis* was confirmed as being the most significant process affected by BDE-47, with the highest Normalized Enrichment Score (NES) as absolute value, with a negative sign since it features entities with the strongest down-regulation (Appendix A). The same for the Hallmark *hypoxia*, enriched only by BDE-47. Interestingly, *fatty acid metabolism* and *mTORC1 signaling* were enriched by BDE-47 and -99 with the same NES negative sign and by MIX with a positive NES, as also evidenced by the heatmaps in Figure 3 (Figure 3a,b). Conversely, the *PI3K/AKT/MTOR signaling* had a positive correlation with BDE-99 and BDE-209 profiles (Figure 3d). BDE-47 was also positively associated with the *KRAS signaling* and negatively with the *reactive oxygen species pathway*, *UV response UP,* and *allograft rejection.*

The top significant Hallmark for the BDE-99 treatment condition was *epithelial–mesenchymal transition*, whereas for BDE-209 the most significant was *Il2/Stat5 signaling*. Interestingly, BDE-209 uniquely enriched other metabolic Hallmarks compared to the other congeners, such as *heme metabolism* and *oxidative phosphorylation*, but also two immune Hallmarks such as *complement* and *inflammatory response*. BDE-209 treatment yielded an opposite correlation for *UV response UP* (NES positive) and *MYC targets V1* (NES negative), compared to BDE-47 and MIX, respectively (Figure 3c,e).

The MIX mostly enriched Hallmarks related to cell cycle progression (*E2F targets, G2M checkpoint, MYC targets V1* and *V2, mitotic spindle)*; also relevant was the *TNFα signaling* via *NFKB.* In addition, we noted the positive correlation with the *estrogen response late* hallmark; on the contrary, BDE-47 had a borderline significance for the *estrogen response early* term (*p* = 0.058; data not shown).

### 2.4. Interaction Networks of lncRNAs Deregulated by PBDEs

Since lncRNAs regulate several processes also by binding to mRNA and proteins [29], we investigated for known validated interactions between identified DELs and corresponding DEGs/DEPs in the same treatment condition, obtaining four different interaction networks (Appendix A). Figure 4 shows networks filtered for the most relevant nodes which included only DELs and DEPs.

The interactome for the BDE-47 condition featured 55 nodes (34 DELs, 11 DEGs, and 10 DEPs) (Appendix A). Main hubs of the network are two induced DELs (*SNHG17* and *LINC01348*) and five induced DEPs (HNRNPA1, NOP58, PRPF8, DHX9, POLR1A) (Figure 4a). 

The interaction network related to the BDE-99 treatment condition included 59 nodes (25 DELs, 25 DEGs, 9 DEPs) (Appendix A). Also in this case, the up-regulated HNRNPA1 and POLR1A proteins were among the top relevant hubs, together with other induced (RANGAP1 and GNL3) and repressed (EIF3B) DEPs, and two induced DELs (*NEAT1* and *RMRP*) (Figure 4b). Interestingly, while *NEAT1* mainly interacted with genes, *RMRP* had more interactions with proteins (Appendix A). 

The interactome related to the BDE-209 treatment featured 75 nodes (25 DELs, 30 DEGs, and 20 DEPs) (Appendix A). Top hubs were represented by four up-regulated DELs (*NEAT1, RMRP*, *SNHG7,* and *BAALC-AS1*), as well as some induced (U2AF2, NOP58 and IGF2BP3) and repressed (HNRNPA1, NUDT21, AIFM1) DEPs (Figure 4c). 

The interaction network obtained for the MIX treatment was the more complex with 219 nodes (61 DELs, 141 DEGs, and 17 DEPs) (Appendix A). Conversely to the previous networks, only down-regulated DELs were included as main interacting hubs (*RAD51-AS1*, *SNHG12, SNHG7, KCNQ1OT1, LINC-PINT, MIR3142HG, AC020916.1, GASAL1, EPB41L4A-AS1, GABPB1-AS1, SNHG17, SCARNA9, PVT1, CCDC18-AS1, TP53TG1, ERVH48-1*) together with induced (U2AF2, NOP58, PRPF8, HNRNPM, TIA1, SND1, DHX9, GNL3, FXR1, RO60, FTO) and repressed (HNRNPA1) DEPs (Figure 4d). Overall, in the MIX condition, *RAD51-AS1* emerged as a main interactor of genes, whereas the other DELs mainly interacted with proteins (Appendix A). 

Overall, some DELs were shared among conditions, such as *NEAT1* and *RMRP* common to BDE-99 and -209 networks, or *SNHG7* and *SNHG17* shared by MIX with BDE-209 and BDE-47, respectively, but with inverse modulations. Interestingly, the protein HNRNPA1, induced by BDE-47 and -99 and repressed by BDE-209 and MIX, was the only hub present in all four networks, interacting with different DELs.

### 2.5. Interaction Networks among DEGs and DEPs and Relevant Hubs

As a next step, we analyzed the most relevant hubs in each interaction network of the four treatment conditions, as shown in Figure 5, also comparing results with previously obtained DELs-DEGs-DEPs networks. Some of the identified hubs are modulated at gene and protein expression levels, being represented as DEPs and placed at the center of each network (Figure 4), except for MIX where all DEPs are also in the DEG list.

Among the 55 most relevant hubs identified in the BDE-47 interaction network (Figure 5a), 7 are modulated at gene and protein expression levels and interacted with 16 DEGs and 32 DEPs. Interestingly, the induced proteins HNRNPA1, NONO, DHX9, and PRPF8 are those mostly interacting with DELs (Figure 4a and Appendix A).

Fifty highly relevant hubs were identified for the BDE-99 interactome (Figure 5b), including 24 DEPs, three features modulated at gene and protein expression level and 23 DEGs. *DDX17,* and *GNAI41* genes both interacted with the lncRNA *NEAT1*; in addition, HNRNPA1 interacted with 21 DELs, including *NEAT1* (Appendix A).

In the BDE-209 interaction network (Figure 5c), we identified 54 relevant hubs featuring 21 DEPs, 4 entities modulated at gene and protein expression level and 29 DEGs. The induced RBM39 interacted with the lncRNA *SNHG7* whereas the repressed PRPF40A interacted with *NEAT1*; moreover, U2AF2, SAFB2, XPO5 and HNRNPA1 interacted with several DELs (Appendix A).

Fifty-nine hubs were obtained in the MIX interaction network, including fourteen entities that were modulated at gene and protein expression levels (indicated as protein hubs in Figure 5d) and forty-five DEGs. Among the identified hubs, some interacted with one lncRNA each, such as *CLIC4* or *HSP90AA1*; besides, four hubs interacted with *RAD51-AS1* only (i.e., *RPL21, SPP1, RPA2* and *UQCRC2*), whereas RBM39, and especially DHX9 and HNRNPA1 interacted with several DELs (Appendix A).

Remarkably, as observed in the DELs interaction networks, the only top hub present in all the four treatment conditions was HNRNPA1.

### 2.6. Real-Time PCR of Selected Markers

On the basis of transcriptome data and functional analysis, we analyzed the gene expression of some selected markers (Figure 6). 

EGFR was significantly up-regulated only by BDE-47 and not by BDE-209; in addition, a borderline significant down-regulation by MIX was noted (*p* = 0.0621). MAPK3 was significantly induced not only by BDE-47, as observed in transcriptomics, but also by MIX and, to a higher extent, by BDE-99.

We also assessed the expression of a panel of nuclear receptors potentially involved in the mode of action of the PBDE congeners (Figure 6). The two ERs were differently modulated by PBDEs, with BDE-47 and BDE-99 significantly up-regulating ERβ while having no effect on ERα. On the contrary, BDE-209 and MIX significantly induced both ERs, especially MIX. HIF-1α was significantly repressed by BDE-99 and -209. BDE-99 was the only congener to significantly repress PPARγ, whereas all the treatment conditions significantly induced PPARα expression.

### 2.7. Glucose Secretion

To verify if the inhibition of the glycolysis pathway induced by BDE-47 was phenotypically evident in HepG2 cells, also in the other treatment conditions, we measured the glucose secretion by cells treated with single PBDE congeners or the MIX. In addition to the 1 nM, we also tested the 10 nM concentration in order to be sure to observe an adverse outcome. 

Effectively, a significant decrease in glucose levels was observed in conditioned medium of cells treated with BDE-47 or -209 only at 10 nM (Figure 7a); a similar, not significant (*p* = 0.0673), decrease was noted also for BDE-99. Interestingly, MIX 1 nM induced a slight not significant (*p* = 0.0649) increase in glucose level.

### 2.8. Oil Red Staining of Intracellular Lipids

As for glucose secretion, we assessed intracellular lipid accumulation in HepG2 cells by the Oil Red staining at both 1 and 10 nM treatment concentrations, to phenotypically link the observed signaling derangement. Results evidenced that all the PBDE congeners and the MIX induced a significant increase in lipid accumulation at both 1 and 10 nM, except BDE-47 at 10 nM (Figure 7b,c). Interestingly, lipid accumulation slightly decreased with the dose of single congeners but not for MIX treatment.

## 3. Discussion

In the present study, we analyzed transcriptomic and proteomic profiles of the human hepatoma cell line HepG2 following treatment with the three most relevant PBDEs congeners for human exposure through diet, i.e., BDE-47, -99, and -209, as well as to their ternary mixture (MIX), at concentrations really occurring in food items [3]. We previously demonstrated that these concentrations did not affect cell vitality and cell proliferation of HepG2 cells [28]. Despite the low concentrations used, all the treatment conditions affected several pathways, although with different modes of action, modulated the expression of some relevant markers and increased intracellular lipid accumulation; in addition, the single congeners alone, but not in the mixture, depleted glucose secretion. 

BDE-47 was the congener affecting the highest number of pathways and the only enriching the *glycolysis* pathway and Hallmark, down-regulating several DEPs involved in almost all the metabolic steps, thus mechanistically supporting an early molecular signaling at 1 nM, which becomes evident at 10 nM with a significant decrease in glucose secretion in HepG2 cells. We observed a similar drop also in cells treated with BDE-209, and marginally also with BDE-99, but no concomitant pathway enrichment was recorded.BDE-47 and -209 significantly enriched also the *fatty acid elongation* pathway; in addition, BDE-47, -99, and the MIX enriched the *fatty acid metabolism* Hallmark, supported by the increased intracellular lipid accumulation in HepG2 cells, also at 1 nM. A number of previous reports described in vivo effects of PBDEs on glucose and lipid metabolism; we limit our discussion to available evidence in liver and blood biomarkers. 

Administration of BDE-47 to adult rats at low doses (0.001–1 mg/kg bw) increased high-fasting glucose in rats with signs of hepatosteatosis and enrichment in pathways related to glucose homeostasis and diabetes [25]. Similarly, treatment of male mice with BDE-209 at doses in the range of 25–75 mg/kg bw, increased the levels of glucose, cholesterol, triglycerides, and low-density lipoproteins (LDL) in the liver, but decreased the levels of glycogen and high-density lipoproteins (HDL) [30]. Increased liver weight, fasting blood glucose, total cholesterol, triglycerides, LDL, and leptin levels as well as decreased serum insulin, HDL, and adiponectin levels were observed in adult male mice treated with BDE-209 at medium-high doses (300–1000 mg/kg bw) [31]. Perinatal exposure to BDE-47 at a low dose (0.2 mg/kg/bw) altered metabolism programming in rat fetuses increasing offspring body weight and glucose uptake [32]. In addition, developmental exposure of male mice to BDE-47 at the same dose misbalanced the blood/liver equilibrium inducing long-lasting effects on triglyceride levels, decreased in the blood, and increased in the liver [33]. A similar outcome was also observed in a human study displaying an inverse association between BDE-99 levels in mothers at delivery and triglyceride levels in children at 6–7 years [23]. 

Overall, in these animal studies the circulating levels of glucose increased upon treatment with PBDEs, thus apparently being in contrast with our results. However, some in vitro studies investigating the mechanism of the possible modes of action underlying these altered metabolic outcomes in liver are in agreement with our results. A transcriptomic study in livers of rats developmentally exposed to low doses of BDE-47 (0.002 and 0.2 mg/kg bw) reported enrichment of the *glycolysis/gluconeogenesis* and *fatty acid metabolism* pathways in association with increased serum cholesterol [34]. Treatment of human embryonic stem cells with BDE-47 or -209 at 10–1000 nM mainly enriched KEGG pathways related to steroid hormones biosynthesis, PPAR signaling and xenobiotic metabolism [35]. Involvement of the PPAR signaling was also observed in hepatic L02 cells treated with BDE-209 at 50 µM which induced PPARγ and mTOR protein expression [30]. Another in vitro transcriptomic study on HepaRG cells treated with BDE-47 and -99 at 25 µM reported enrichment of the *PPARα-RXRα* and the *mTOR pathways* as well as a downregulation of pathways related to the carbohydrate metabolism [36]. Moreover, activation of the mTORC1 signaling has been observed following administration of BDE-47 in HepG2 cells (1 µM) and in the liver of mice (0.2 mg/kg bw), with long-lasting effects on triglyceride levels [37]. 

In our experimental conditions, the *mTORC1 signaling* Hallmark was negatively associated with BDE-47 and -99 DEGs/DEPs and positively associated with MIX treatment. Further, both BDE-99 and -209 were positively associated with *PI3K/AKT/mTOR signaling*. Following activation by growth factors, the PI3K/AKT signaling triggers the formation of one of the two possible mTOR complexes, mTORC1, which is the main regulator of cell growth and differentiation, protein synthesis, and ribosomal biogenesis but it also controls lipid and glucose metabolism by trans-activating different transcription factors [38]. In particular, mTORC1 activates PPARγ but inhibits PPARα, thus promoting fatty acid synthesis and repressing fatty acid oxidation and ketone bodies synthesis, respectively [38]. We did not observe any PPARγ induction, not even target genes; rather, BDE-99 significantly down-regulated its gene expression. On the other hand, we recorded a strong induction of PPARα gene expression by all single congeners and by MIX to a greater extent, thus confirming what was observed in HepaRG cells with concomitant PPARα and mTOR activation [36]. However, lipogenesis may be induced by mTOR also through the trans-activation of sterol-regulatory-element-binding proteins (SREBPs) [38], as observed in marine copepod administered with BDE-47 [39]. We found that a protein encoded by a target gene, e.g., lanosterol synthase (LSS), was induced by BDE-47, -209, and the MIX, which also induced other targets down-stream SREBPs signaling, e.g., fatty acid synthase (FASN) and ATP citrate lyase (ACLY) proteins [38], thus letting assume that the observed lipid accumulation may have been determined by activation of the SREBPs signaling rather than PPARγ one. Interestingly, we previously observed an increase in ATP production induced by BDE-47, -99, and -209 in HepG2 cells at 10 nM [28], which may result in a depletion of adenosine monophosphate–activated protein kinase (AMPK) levels, with consequent induction of lipogenesis, by SREBPs activation, and glycolysis reduction [40]; thus, previous evidence further substantiate the present results.

BDE-47 treatment was the only to enrich the *HIF-1 signaling pathway* and the *hypoxia* Hallmark but did not significantly modulate HIF-1α gene expression; conversely, BDE-99 and -209 significantly repressed its expression, thus also these congeners may possibly affect such pathways, but maybe to a minor extent since no significant enrichment of the related pathway was found. Noteworthy, mTORC1 interacts also with HIF-1α [41], thus participating in the regulation of transcription in hypoxic conditions. The *HIF-1 signaling pathway* was highly interconnected with *glycolysis* due to the sharing of several repressed DEPs of the anaerobic metabolism such as enolases (ENO1, ENO2, ENO3) and the glyceraldehyde 3-phosphate (GAPDH), lactate (LDHB) and pyruvate (PDHB) dehydrogenases. Limited evidence is available on PBDE’s effects on HIF-1α. A previous report described the inhibition of *hif-1a* expression in a fish fibroblast cell line upon treatment with BDE-47 and BDE-99 at 100 µM [42]. Conversely, BDE-47 and -209, as well as the ternary mixture at 1 µM did not affect HIF-1α protein expression in human foreskin fibroblast HS-68 cells, while BDE-99 strongly increased it [43], thus suggesting that modulation may be different according to the cellular type.

In HepG2 cells treated with BDE-47, the *glycolysis* pathway was also highly interconnected with the *estrogen signaling pathways*; in addition, BDE-47 determined a borderline significance for the *estrogen response early* Hallmark, whereas the MIX was positively associate with the *estrogen response late* Hallmark. No imbalance of estrogen signaling was evidenced by functional analysis for BDE-99 and -209 treatments. Both BDE-47 and BDE-99 up-regulated the gene expression of ERβ but not ERα, whereas BDE-209 significantly increased the expression of both ERs. Interestingly, the MIX markedly increased ERs expression, thus supporting some additive effect. Several PBDEs were demonstrated to bind and act as weak agonists of ERα, including BDE-47 [12]. However, the influence of BDE-47 on ERs expression appeared to vary with the cell type; indeed, in MCF7 breast cancer cells, BDE-47 decreased ERα and increased ERβ protein expression at 5 ng/mL whereas it had no effect on OVCAR-3 ovarian carcinoma cells [44]. Using reporter gene assays, the methoxylated metabolite of BDE-47 displayed estrogenic activity at 10 μM and anti-estrogenicity at higher concentrations [45]. To our knowledge, no previous report on ERs gene expression modulation in HepG2 following treatment with PBDE is available.

In the pathway networks of PBDE-treated HepG2 cells, *EGFR* appeared at the crossroad of *HIF-1* and *estrogen signaling* for BDE-47 and was shared by *prostate cancer* and *adherens junction* pathways for BDE-209. EGFR is a membrane receptor of the tyrosine kinase family mainly controlling cell proliferation and development, thus being involved in cancer pathogenesis, but also regulating apoptosis, oxidative status, and metabolism [46]. Both BDE-47 and -209 upregulated *EGFR* in transcriptomics, but only for BDE-47, we confirmed its gene expression induction by qPCR. Upon binding of a growth factor, EGFR triggers the activation of several signaling cascades, including the MAPK signaling pathway, whose last effectors are ERK1/2, also known as MAPK3 and MAPK1, respectively. BDE-47 increased the gene expression of *MAPK3* and decreased the protein expression of MAPK1, but their function is redundant [47], so a balance may occur. On the contrary, BDE-209 unaffected *MAPK3* expression while inducing MAPK1; BDE-99 strongly increased *MAPK3* expression (by qPCR) with no effect on MAPK1, thus different mechanisms may be hypothesized for the three congeners. In addition, the MIX did not affect MAPK1 and *MAPK3* in omics, but this last was found to increase qPCR. These findings are mostly supported by a previous report in HS-68 human fibroblasts showing increased ERK1 (MAPK3) protein expression after exposure with 1 µM BDE-47, -99, -209 and their ternary mixture [43]. However, in our experimental conditions, the modulations were not sufficiently striking to significantly derange the above-mentioned pathways, except in HepG2 cells treated with BDE-47.

Noteworthy, EGFR and ERs have a mutual dialogue by activating each other; indeed, after EGFR stimulation, MAPKs (ERKs) phosphorylate ERα thus determining the transactivation of ERα-target genes whereas the inverse signaling from ERs to EGFR occurs through a rapid membrane cross-talk [48]. Since BDE-47 was borderline significant for the *estrogen response early* Hallmark, this may possibly support a rapid membrane estrogen activation by BDE-47 through ERβ, then triggering EGFR cascade. A previous report described the involvement of both ERα and the EGFR/ERK signaling pathways in Ishikawa cells treated with BDE-47 at 10 µM [49], thus confirming the concomitant induction of both kinds of receptors by BDE-47.

To our knowledge, the cumulative derangement of PI3K/AKT/mTOR, HIF-1α, ERs, and EGFR/MAPK signaling have never been described in relation to PBDE exposure in general and BDE-47 in particular. However, the gathered evidence on single pathways activation lends support to our findings.

By the transcriptomic analysis, we identified several lncRNAs modulated by PBDEs, some of which are involved in metabolic pathways and appeared to have central roles in the interactome of PBDE-treated HepG2 cells. BDE-47 induced the expression of *SNHG17* and *LINC01348*, respectively, an oncogene and a tumor suppressor in hepatocellular carcinoma (HCC) [50,51]. As such, they have opposing activities on cell proliferation and migration, the first promoting and the second mitigating such processes [50,51]. Thus, their concurrent induction by BDE-47 may result in a balance of these actions. Interestingly, *SNHG17* mostly affected metabolic pathways in HCC [50]. 

In the BDE-99 and -209 interaction networks with DELs, *RMRP* and *NEAT1* were top up-regulated hubs. Both are negatively correlated with prognosis of patients with HCC [52,53]. In addition, *RMRP* was recently demonstrated to be up-regulated in liver of patients affected by non-alcoholic fatty liver disease (NAFLD); its inhibition decreased steatosis and lipid deposition in a NAFLD rat model [54]. Similarly, also *NEAT1* has a central role in NAFLD, promoting disease progression [53]. Its suppression restored excessive lipid metabolism, and repressed inflammation, apoptosis and liver fibrosis [53,55]. Also the up-regulated *SNHG7* and *BAALC-AS1* were top DELs interactors in the BDE-209 network, and both are involved in cancer progression, including HCC, promoting cell proliferation and suppressing apoptosis [56,57].

The 17 DELs identified as top hubs in the MIX interaction network were all repressed by the combined action of the three single congeners, including *SNHG7* and *SNHG17*, induced by BDE-209 and BDE-47. Among the other relevant DELs, *EPB41L4A-AS1* was found to be regulated by p53; its low expression in most cancer types seems to be related to the Warburg effect with increased aerobic glycolysis [58]. Similarly, the top interacting *RAD51-AS1* was found down-regulated in colon adenocarcinoma and colorectal cancer patients; induced accumulation of *RAD51-AS1* mitigated cell invasion and migration as well as glucose uptake [59]. Thus, its repression by the ternary mixtures of PBDE may contribute to the opposite effects. In addition, *LINC-PINT* expression, found to be repressed in hepatocytes of hepatitis C-affected individuals, was responsible for lipogenesis enhancement [60].

From the DELs validated interaction networks as well as from the DEGs/DEPs interaction networks, we found that HNRNPA1 always represented a central hub, being induced by BDE-47 and -99 and repressed by BDE-209 and MIX. HNRNPA1 is a splicing factor with tissue-specific activity [61] and only one report described its induction by BDE-209 in Sertoli cells of mice administered with 0.25 mg/kg bw as splicing factor of the thyroid receptor alpha 2 [62]. However, HNRNPA1 was recently demonstrated to promote the translation of genes related to fatty acid oxidation upon interaction with the lncRNA H19 [63]. Further, HNRNPA1 expression participates in the regulation of pyruvate kinase and glucose metabolism following activation by c-Myc [64]. Importantly, *Myc targets* Hallmark was significantly enriched in HepG2 treated with BDE-209 and MIX. Thus, its presence as a central hub in the networks supports its main role as a regulator of glucose and lipid metabolism possibly by interacting with DELs and transcription factors such as c-Myc. Interestingly, it has been reported the role of EGFR as a regulator of alternative splicing, promoting the transcription of HNRNPA1 through the MAPK signaling cascade, also in HepG2 cells, finally leading to Insulin receptor upregulation [65]. In addition, activation of SREBP-1a by HNRNPA1 has been observed in HepG2 cells and primary rat hepatocytes in response to endoplasmatic reticulum stress [66]. Therefore, HNRNPA1 seems to represent a critical marker of PBDEs exposure being related to most of the signaling pathways deranged by these chemical compounds, also interacting with DELs.

Overall, the integrated analysis of transcriptomic and proteomic data demonstrated to more extensively elucidate the contribution of deregulated genes and proteins in the affected pathways and Hallmarks; moreover, considering the interacting role of modulated lncRNAs. By this approach, the present study provided mechanistic evidence of BDE-47, -99, and -209 different modes of action involving several inter-connected pathways. BDE-47 was the congener affecting the higher number of pathways and biological processes, followed by BDE-209 and -99. The ternary mixture did not display an obvious additive effect, rather being affected in the opposite manner compared to the single congeners, possibly because of the different mechanisms triggered by each chemical compound. On the basis of this evidence, more targeted studies could enhance the knowledge of the interaction and contribution of single PBDE congeners in mixture, especially at human-relevant concentrations, thus improving their risk assessment.

## 4. Materials and Methods

### 4.1. Chemicals

PBDE congeners 47, 99 and 209 were purchased by Wellington Laboratories (Guelph, ON, Canada). Solvent was eliminated by nitrogen flushing and the dry powder was dissolved in dimethyl sulfoxide (DMSO) to obtain 5 µg/mL standard solutions stored at 4 °C. Working dilutions were prepared fresh before cell treatment in culture medium.

### 4.2. Cell Treatment

HepG2 cell line was purchased by ATCC (Manassas, VI, USA) and grown in DMEM/F12 medium without phenol red (Life Technologies, Paisley, UK), supplemented with 10% fetal bovine serum (Lonza, Basel, Switzerland), 2 mM L-glutamine (Lonza), 100 U/mL penicillin and 100 mg/mL streptomycin (Life Technologies). Cells were maintained in a humidified Steri-Cult 200 Incubator (Forma Scientific, Marietta, OH, USA) at 37 °C and 5% CO_2_. Based on previous results on HepG2 and median chronic intake in the general population [3,28], the 1 nM concentration was chosen for each congener to treat cells. The equimolar mixture of the three congeners, each at 1 nM, was also prepared to mimic a realistic median co-exposure of these PBDEs by food intake.

Cells were plated and incubated overnight at 37 °C, then medium was replaced with fresh medium containing BDE-47, -99, or -209 congeners at 1 nM concentration, the equimolar mixture of the three congeners at 1 nM concentration (MIX), or medium alone as control, incubating cells for 72 h at 37 °C. Since dilution from the standard solution ranged from over 1:5000 (for BDE-209) to over 1:10.000 (for BDE-47), we did not include a control with DMSO being 0.04% at the highest in the MIX, thus far below the toxicity threshold [67]. At the end of incubation, cells were harvested, centrifuged, and cell pellets stored at −80°C until further processing. Three independent experiments were performed for both transcriptomic and proteomic analyses.

### 4.3. Transcriptomic Analysis

HepG2 cell pellets dedicated to transcriptomic analysis were extracted for their total RNA content by the Norgen kit (Norgen Biotek Corp., Thorold, ON, Canada). RNA quantity was assessed by NanoDrop spectrophotometer reading (Thermo Fisher, Waltham, MA, USA), and integrity by agarose gel electrophoresis. All the samples met quality criteria to be further processed.

Samples preparation for microarray analysis was previously described [68,69]. Briefly, from 1 µg of each RNA sample, we performed cDNA synthesis, cRNA amplification, and labeling by the Quick Amp Labeling Kit Two Colors (Agilent, Santa Clara, CA, USA). Samples were alternatively labeled with Cy5 and Cy3 and co-hybridized with a dye swap protocol on 4 × 44 K v2 whole human genome microarray slides (Agilent) at 65 °C for 17 h, with a 10 rpm rotation in an Agilent hybridization oven. Slides were washed and scanned by the microarray scanner system (Agilent), extracting data through the Feature Extraction 9.5 software (Agilent).

Differential expression analysis was performed with *limma* 3.48 [70] in R 4.0 [71]. Data were normalized within each array by a global LOESS and between arrays by quantile normalization. Signals from the two channels were transformed into single-channel data calculating the correlation between channels for the same spot. Differential gene expression analysis for each treated group vs. control samples was performed applying a Bayesian model, setting significance at *p* < 0.01 and controlling for the false discovery rate (FRD) set < 5%. Since some transcripts were classified as unknown, especially the long noncoding transcripts (lncRNAs), we updated the annotations using the bioMart package 2.48 [72] in R according to the genome assembly GRCh38.p12 release 98. A Venn diagram for the visualization of overlapping modulated genes across conditions was generated with Venny 2.1 [73]. 

Raw and normalized microarray data have been deposited in NCBI’s Gene Eexpression Omnibus and are accessible through GEO Series accession number GSE216590 (https://www.ncbi.nlm.nih.gov/geo/query/acc.cgi?acc=GSE216590, accessed on 28 October 2022).

### 4.4. Proteomic Analysis

HepG2 cell pellets dedicated to proteomic analysis were homogenized in RIPA buffer (50 mM Tris-HCl, pH 7.5; 150 mM NaCl; 1% Triton X-100; 1% sodium deoxycholate; 0.1% SDS) containing a protease inhibitor cocktail (Sigma-Aldrich, Castle Hill, NSW, Australian) on ice. Protein lysates were clarified by centrifugation at 14,000 rpm for 15 min at 4 °C. Total protein content in each sample was determined by the Pierce BCA Protein Assay Kit (Thermo Scientific, Rockford, IL, USA) according to manufacturer’s instructions, reading absorbance at 562 nm on a NanoDrop spectrophotometer (Thermo Fisher). Protein concentrations were extrapolated from a standard curve of bovine serum albumin serial concentrations by the GraphPad Prism 5.0 software (Graph-Pad Software Inc., La Jolla, CA, USA). 

An amount of 25 µg of total proteins from each sample was loaded and separated on 1D-gel NuPAGE 4–12% (Invitrogen, Carlsbad, CA, USA) run in MOPS buffer (Invitrogen) and stained with the colloidal blue staining kit (Invitrogen). Each gel lane was cut into 12 homogeneous slices and subjected to trypsin digestion after reduction and alkylation steps [74]. The resulting peptide mixtures were separated by a nanoflow reversed-phase liquid chromatography–tandem mass spectrometry using an HPLC Ultimate 3000 (DIONEX, Sunnyvale, CA, USA) connected with a linear ion trap (LTQ-XL, ThermoElectron, San Jose, CA, USA) equipped with a nano-electrospray ion source (ESI). 

Peptides were desalted in a trap column (Acclaim PepMap 100 C18, LC Packings, DIONEX) and separated in a 10 cm long reverse phase column (Silica Tips FS 360-75-8, New Objective, Woburn, MA, USA) slurry-packed in-house with 5 μm, 200 Å pore size C18 resin (Michrom BioResources, Auburn, CA, USA). Peptides were eluted using a linear gradient from 96% A solution (H_2_O with 5% acetonitrile and 0.1% formic acid) to 60% B solution (acetonitrile with 5% H_2_O and 0.1% formic acid) for 40 min, at 300 nL/min flow rate.

Peptide ions were analyzed in positive mode, and the high voltage potential was set up at around 1.7 kV. Full MS spectra (*m*/*z* 400–2000 mass range) were acquired in a data-dependent mode in which each full MS scan was followed by five MS/MS scans; the five most abundant molecular ions were dynamically selected and fragmented by collision-induced dissociation, using a normalized collision energy of 35%. A set-up of pre-column and analytical column was used.

Database searches were performed using the Sequest algorithm, embedded in Proteome Discoverer 1.4 (PD; Thermo Fisher Scientific). The search criteria for protein identification were set to match at least two peptides per protein. Tandem mass spectra were analyzed by Sequest HT and matched against the SWISS-PROT human database (release 2016, containing 20116 sequences). Spectra files were searched using PD with carbamidomethylation on cysteine as static modification, methionine oxidation as variable one, and full tryptic peptides with a maximum of two missed cleavages allowed were considered for identification, applying a mass tolerance of 1.5 Da for precursor ion and 0.8 Da for fragment ions.

The Percolator tool within PD was used for peptide validation based on q-values, corresponding to FDR ≤ 1% at peptide-level. For each of the three treatment replicas, two technical replicas were performed. 

Obtained peptide spectrum match (PSM) data were corrected for batch effects by the RUVSeq tool 1.18.0 in R environment, applying the RUVr method and setting k = 10 [75]. Differential expression analysis was performed with edgeR 3.26.8 [76] using the negative binomial GLM analysis and controlling for the FDR (<5%). A Venn diagram for the visualization of overlapping modulated proteins across conditions was generated with Venny 2.1 [73].

Proteomics data have been deposited in MassIVE and are accessible through the identifier MSV000088230 (https://massive.ucsd.edu/ProteoSAFe/dataset.jsp?task=0036e8ca344a49acbc369ee1dba6d527, accessed on 25 October 2022).

### 4.5. LncRNA Interaction Analysis

Validated interactions between differentially expressed lncRNAs (DELs) and genes (DEGs) and/or proteins (DEPs) for each treatment condition were retrieved by the RNAInter database [77], setting the score > 0.5, and the ENCORI database [78]. Results were filtered for genes and proteins present in the DEG and DEP lists, respectively. Cytoscape 3.8 [79] was used to visualize the obtained networks and calculate the topological characteristics by the network analyzer tool [80]. To identify the more relevant hubs, network size was reduced by filtering nodes with betweenness centrality (BC), closeness centrality (CC), and degree (D) values above the 90th percentile of their distribution.

### 4.6. Functional Analysis

The combined DEG and DEP lists for each experimental condition were analyzed for their functional enrichment by two different approaches, namely an over-representation analysis and a Gene Set Enrichment Analysis (GSEA). For the over-representation analysis, assuming independence between queried entities, the combined DEGs and DEPs list of each condition were uploaded into the Cytoscape plugin ClueGO 2.5.7 [81] in four clusters (up and down-regulated DEPs, up and down-regulated DEGs) to analyze the enriched KEGG pathways and visualize the contribution of each cluster, setting *p*-value of the Fisher’s exact test < 0.05. 

For the GSEA analysis, the list of DEGs + DEPs for each condition was ranked from higher to lower values after multiplying the log_2_(FC) × -log_10_(*p*-value) to take into account both the magnitude of modulation and the level of significance. The ranked lists were analyzed by the GSEA 4.3.1 software to identify the most enriched Hallmarks applying the classical algorithm based on the Kolmogorov–Smirnov test, performing 1000 permutations. Significant Hallmarks were selected as having *p*-value < 0.06 and FDR < 25%.

The ggplot2 package v.3.3.6 [82] in the R environment was used to visualize the enriched KEGG pathways and GSEA Hallmarks in all the conditions.

### 4.7. Interaction Network Analysis

Each DEG and DEP list of the different conditions was queried into Cytoscape to retrieve genetic and physical interactions by GeneMania 3.5.2 app and protein–protein interactions by STRING 1.7.0 app, then merging the results. For each condition, topological characteristics were calculated with the Network Analyzer tool. To identify the more relevant hubs, network size was reduced by filtering nodes with BC, CC, and D being above the 90th percentile of their distribution. More stringent filtering was applied to MIX interaction network due to the high number of DEGs and DEPs, considering the 95th percentile.

### 4.8. Real-Time PCR

An amount of 1 µg of each RNA sample was reverse-transcribed by the SensiFast™ cDNA Synthesis Kit (Bioline Reagents Ltd., London, UK). Specific primers for the following selected genes were designed by Primer-BLAST (www.ncbi.nlm.nih.gov/tools/primer-blast, accessed on 30 May 2022) and purchased from Thermo Fisher: EGFR (fw GACAGGCCACCTCGTCG, rev TCGTGCCTTGGCAAACTTTC); MAPK3 (fw GGGCATCCTGGGCTCCCCAT, rev CGGCCACTGGCTCATCCGTC); HIF1α (fw GGACAGCCTCACCAAACAGA, rev GATTGCCCCAGCAGTCTACA); ERα (fw TGGGAATGATGAAAGGTGGGAT, rev GGTTGGCAGCTCTCATGTCT); ERβ (fw TGAAACTTGCAGGGCGAAGA, rev GGGCAAGTATAATGGCTTGCAG); PPARγ (fw GATGACAGCGACTTGGCAAT, rev AGGAGCGGGTGAAGACTCAT); PPARα (fw ACCACAGTTCTGGAGGCTGGGA, rev GCCTCGAGTGGGGAGAGGGG) and RPL13a as housekeeping gene (fw CGCCTGGCTCACGAGGTTGG, rev GGAGGAAGGGCAGGCAACGC). The Excel TaqTM Fast Q-PCR Master Mix SYBR (SMOBIO Technology Inc., Hsinchu City, Taiwan) was used to prepare PCR reactions which were run in duplicate in a Bioer LineGene 9600 Plus thermocycler instrument (Bioer Technology Co. Ltd., Hangzhou, China) with the following thermal program: 1 cycle at 95 °C for 20 s; 40 cycles at 95 °C for 3 s, 58 °C for 15 s and 72 °C for 15 s; 1 melting cycle from 55 to 95 °C, 30 s/°C. Using the LineGene 9600 PCR V.1.0 software (Bioer), we obtained the threshold cycles (Ct) in each condition and calculated ΔΔCt values, with control cells as calibrators and RPL13a as normalizer.

### 4.9. Glucose Assay

The Glucose Colorimetric Detection Kit (Thermo Fisher) was used following manufacturer’s protocol. Briefly, 10,000 cells/well were seeded on 96 flat-bottomed multi-wells and incubated overnight at 37 °C. In order to clearly detect an adverse outcome, the medium was replaced with fresh medium containing 1 nM and also a higher concentration, 10 nM, of BDE-47, -99 -209, their equimolar ternary mixture (MIX), or medium alone as control, incubating cells for 72 h at 37 °C. After treatment, conditioned medium in each well was collected and diluted 1:20 in assay buffer. An amount of 20 µL of diluted samples or the provided glucose standard (0.5–32 mg/dL) were transferred to a new 96 flat-bottomed multi-wells and added with horseradish peroxidase, the substrate and glucose oxidase, incubating for 30 min at RT. Absorbance was read at 560 nm by a Victor 3 Multilabel Reader (PerkinElmer, MA, USA). Unknown concentration in each sample was derived by the glucose standard curve designed with GraphPad Prim 5.0. The assay was repeated in three independent experiments.

### 4.10. Oil Red O Assay

The Lipid (Oil Red O) Staining Kit (BioVision, Milpitas, CA, USA) was used following the manufacturer’s protocol. Briefly, 150,000 cells were seeded in 24 flat-bottomed multi-wells and incubated overnight at 37 °C. As for glucose assay, to clearly detect an adverse outcome medium was replaced with fresh medium containing 1 nM and also 10 nM concentrations of BDE-47, -99 -209, their equimolar ternary mixture (MIX), or medium alone as control, incubating cells for 72 h at 37 °C. After treatment, the medium was removed, and cells were washed with PBS and fixed with 10% formalin for 30 min. Cells were then stained for 15 min with Oil Red O working solution and 1 min with hematoxylin. After washing with dH_2_O, cells were observed by an inverted microscope (Nikon Eclipse Ts2, Nikon Europe B.V. Amstelveen, The Netherlands) acquiring images with a camera. The cells were then washed twice with 60% isopropanol and once with 100% isopropanol, each time for 5 min with gentle rocking, to dissolve Oil Red from cells. Absorbance at 492 nm was read by Victor 3 Multilabel Reader. The assay was repeated in three independent experiments.

### 4.11. Statistical Analysis

Statistical differences of treatment groups from control cells were calculated with STATA 14.2 (StataCorp, College Station, TX, USA) by the analysis of variance (ANOVA) followed by the post hoc Dunnett’s test where applicable. Correlation analysis between common DEGs and DEPs for each treatment condition was calculated by Pearson’s test. The significance was set at *p* < 0.05 for both tests.

## Figures and Tables

**Figure 1 ijms-23-14465-f001:**
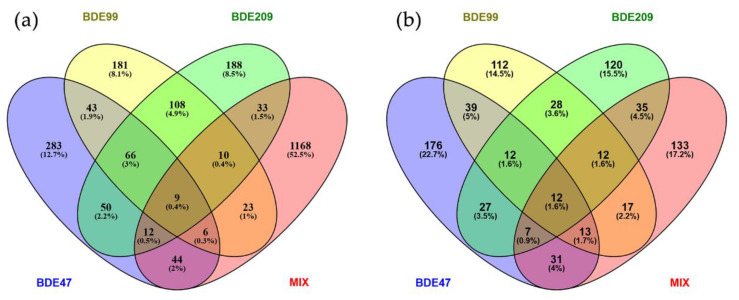
Venn diagrams of the (**a**) genes and (**b**) proteins differentially expressed in HepG2 following treatment for 72 h with BDE-47, -99, -209 and their ternary mixture (MIX) at 1 nM.

**Figure 2 ijms-23-14465-f002:**
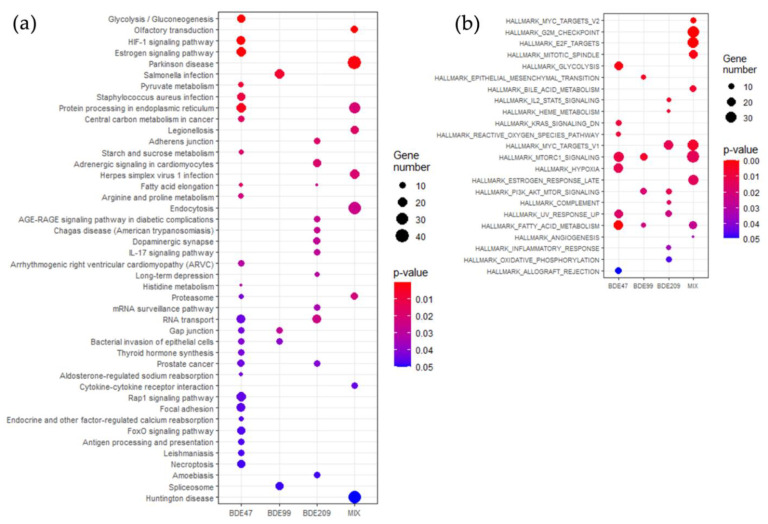
Dot plots of significantly enriched (**a**) KEGG pathways and (**b**) GSEA hallmark gene sets in HepG2 cells treated for 72 h with BDE-47, BDE-99, BDE-209, or MIX at 1 nM. The size of the dots is proportional to the number of featured entities and the color varies according to the *p*-value.

**Figure 3 ijms-23-14465-f003:**
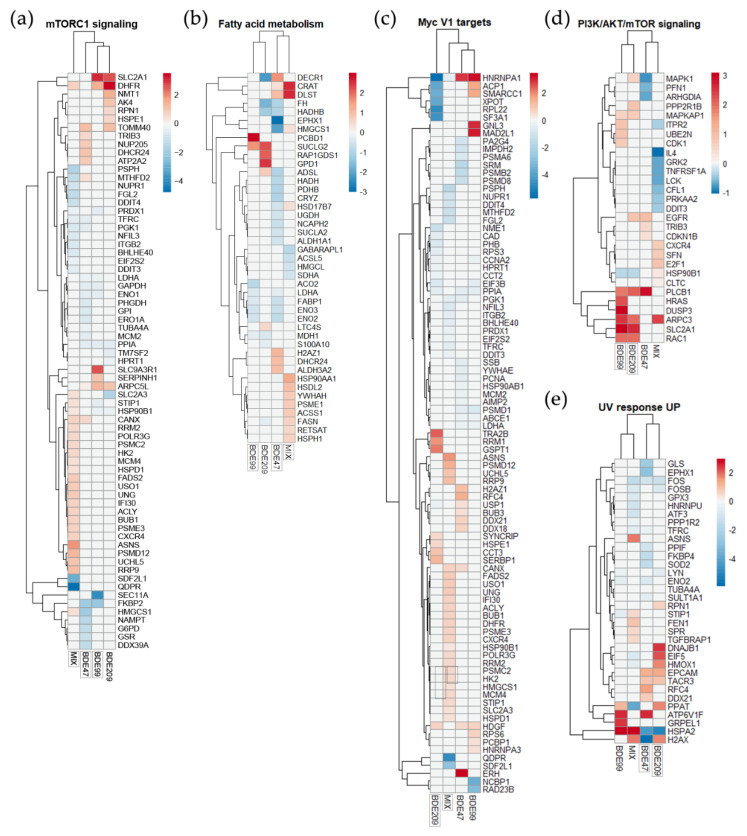
Heatmaps of combined DEGs and DEPs associated with the significantly enriched Hallmark gene sets (**a**) mTORC1 signaling, (**b**) fatty acid metabolism, (**c**) Myc V1 targets, (**d**) PI3K/AKT/mTOR signaling, and (**e**) UV response UP, in HepG2 treated for 72 h with BDE-47, -99, -209 and their ternary mixture (MIX) at 1 nM. Frames around names indicate which congener/MIX treatment significantly enriched the Hallmarks.

**Figure 4 ijms-23-14465-f004:**
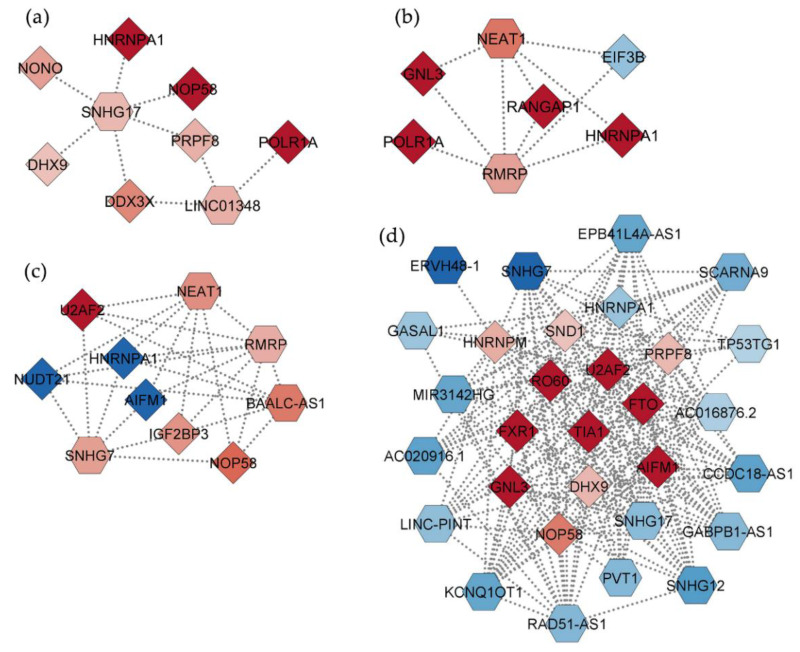
Interaction networks between DELs (hexagons) and DEPs (diamonds), as inferred by publicly available validated experiments, in HepG2 cells treated for 72 h with (**a**) BDE-47, (**b**) BDE-99, (**c**) BDE-209 or (**d**) MIX at 1 nM. Only nodes with BC, CC, and D > 90th percentile are shown. Shades of blue and red represent, respectively, the magnitude of down- or up-regulation.

**Figure 5 ijms-23-14465-f005:**
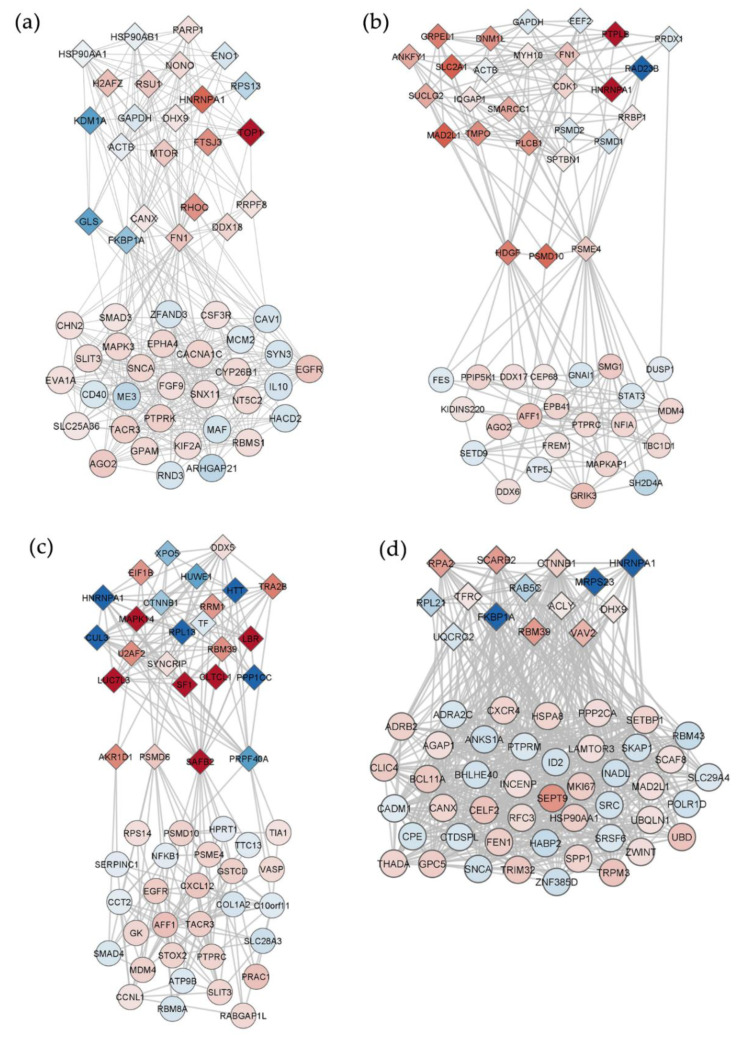
Interaction networks between DEGs (circles) and DEPs (diamonds) in HepG2 cells treated for 72 h with (**a**) BDE-47, (**b**) BDE-99, (**c**) BDE-209 or (**d**) MIX at 1 nM. Only nodes with BC, CC, and D > 90th percentile are shown, except for MIX where the threshold was the 95th percentile. Shades of blue and red represent, respectively, the magnitude of down- or upregulation.

**Figure 6 ijms-23-14465-f006:**
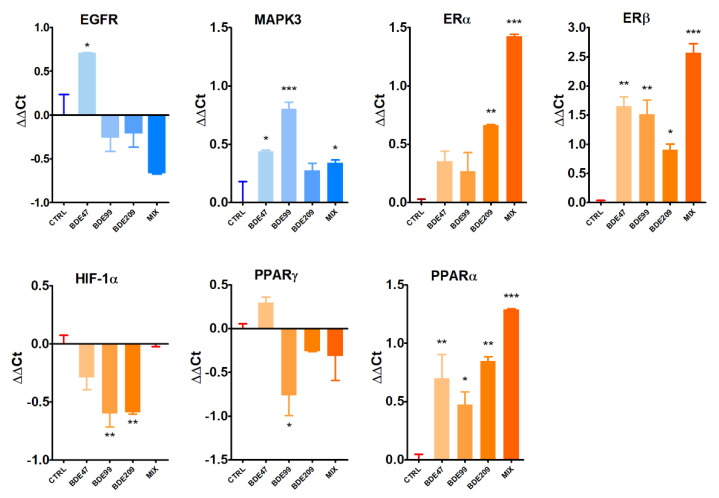
Gene expression analysis by qPCR of selected genes in HepG2 cells treated with BDE-47, BDE-99, BDE-209, or their ternary mixture (MIX) at 1 nM for 72 h. Asterisks indicate the level of significance: * *p* < 0.05, ** *p* < 0.01, *** *p* < 0.001.

**Figure 7 ijms-23-14465-f007:**
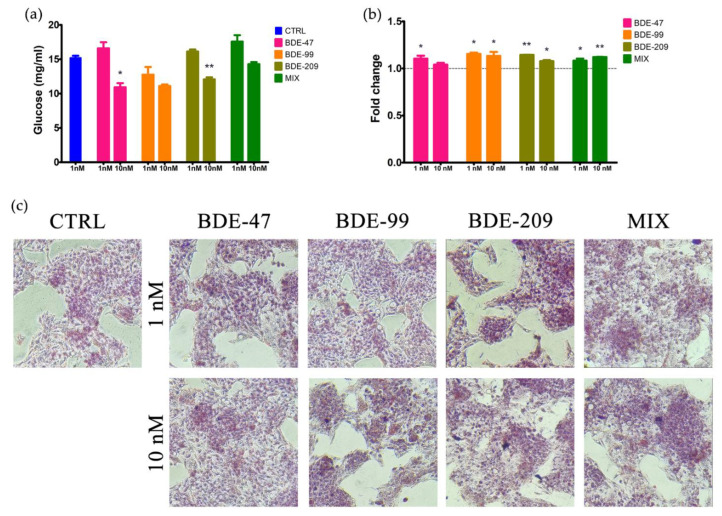
(**a**) Glucose levels measured in conditioned medium of HepG2 cells treated for 72 h with BDE-47, -99, -209 or their ternary mixture (MIX) at 1 and 10 nM. (**b**) Intracellular lipid accumulation detected by fluorescence reading of oil red staining in HepG2 treated as described in (**a**). (**c**) Representative images of HepG2 oil red staining in all the experimental conditions (40× captured images on a Nikon Eclipse Ts2, Nikon Europe B.V. Amstelveen, The Netherlands). Asterisks indicate the level of significance: * *p* < 0.05; ** *p* < 0.01.

**Table 1 ijms-23-14465-t001:** Number of differentially expressed genes (DEGs) and proteins (DEPs) identified in HepG2 cells following treatment for 72 h with BDE-47, -99, -209 and their ternary mixture (MIX) at 1 nM.

	BDE-47			BDE-99			BDE-209			MIX		
**DEGs**	up	down	tot	up	down	tot	up	down	tot	up	down	tot
	309	204	513	324	122	446	330	146	476	446	860	1306
**DEPs**	up	down	tot	up	down	tot	up	down	tot	up	down	tot
	169	148	317	190	55	245	154	99	253	203	58	261

## Data Availability

Raw and normalized microarray data have been deposited in NCBI’s Gene Expression Omnibus and are accessible through GEO Series accession number GSE216590 (https://www.ncbi.nlm.nih.gov/geo/query/acc.cgi?acc=GSE216590, accessed on 28 October 2022). Proteomics data were deposited in the MassIVE and are accessible through the identifier MSV000088230 (https://massive.ucsd.edu/ProteoSAFe/dataset.jsp?task=0036e8ca344a49acbc369ee1dba6d527, accessed on 25 October 2022).

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
