# Peer review of "BDE-47, -99, -209 and Their Ternary Mixture Disrupt Glucose and Lipid Metabolism of Hepg2 Cells at Dietary Relevant Concentrations: Mechanistic Insight through Integrated Transcriptomics and Proteomics Analysis"

_ijms, 2022, doi:10.3390/ijms232214465_

Round 1

Author Response

Dear Reviewer,

please see the attachment for our point by point answers.

Regards

Reviewer 2 Report

The authors tested the effects of BDE-47, -99 and -209 on HepG2 cells through integrated transcriptomics and proteomics analysis, and found that molecules disrupted glucose and lipid metabolism. The manuscript was well written, and the results were well presented. I only have some minor concerns as listed below.

1. I suggest the authors to include some biological experiments to provide more solid evidence. For example, the cell viability after treatments should be provided. The glucose and lipid uptake or oxidation after treatments should be detected. In addition, the protein level was suggested to be detected using WB in Figure 6.

2. Title for each figure should be provided.

Author Response

(The authors gave the same response as above.)
